# FodFoM: Fake Outlier Data by Foundation Models Creates Stronger Visual Out-of-Distribution Detector

### Jiankang Chen
School of Computer
Science and Engineering
Sun Yat-sen University
Guangzhou, Guangdong
China
Peng Cheng Laboratory
Shenzhen, Guangdong
China
Key Laboratory of Machine
Intelligence and Advanced
Computing
Guangzhou, Guangdong
China
chenjk36@mail2.sysu.edu.cn

### Ling Deng
### Zhiyong Gan
denglingl@chinaunicom.cn
ganzy1@chinaunicom.cn
China United Network
Communications
Corporation Limited
Guangdong Branch
Guangzhou, Guangdong
China

### Wei-Shi Zheng
School of Computer
Science and Engineering
Key Laboratory of Machine
Intelligence and Advanced
Computing
Sun Yat-sen University
Guangzhou, Guangdong
China
wszheng@ieee.org

### Ruixuan Wang*
School of Computer
Science and Engineering
Sun Yat-sen University
Guangzhou, Guangdong
China
Peng Cheng Laboratory
Shenzhen, Guangdong
China
Key Laboratory of Machine
Intelligence and Advanced
Computing
Guangzhou, Guangdong
China
wangruix5@mail.sysu.edu.cn

## Abstract

Out-of-Distribution (OOD) detection is crucial when deploying machine learning models in open-world applications. The core challenge in OOD detection is mitigating the model's overconfidence on OOD data. While recent methods using auxiliary outlier datasets or synthesizing outlier features have shown promising OOD detection performance, they are limited due to costly data collection or simplified assumptions. In this paper, we propose a novel OOD detection framework FodFoM that innovatively combines multiple foundation models to generate two types of challenging fake outlier images for classifier training. The first type is based on BLIP-2's image captioning capability, CLIP's vision-language knowledge, and Stable Diffusion's image generation ability. Jointly utilizing these foundation models constructs fake outlier images which are semantically similar to but different from in-distribution (ID) images. For the second type, GroundingDINO's object detection ability is utilized to help construct pure background images by blurring foreground ID objects in ID images. The proposed framework can be flexibly combined with multiple existing OOD detection methods. Extensive empirical evaluations show that image classifiers with the help of constructed fake images can more accurately differentiate real OOD images from ID ones. New state-of-the-art OOD detection performance is achieved on multiple benchmarks. The code is available at https://github.com/Cverchen/ACMMM2024-FodFoM.

*Corresponding author.

## CCS Concepts

- **Computing methodologies** → **Computer vision**.

## Keywords

Out-of-Distribution Detection, Foundation Models, Fake OOD Image Generation

**ACM Reference Format:**
Jiankang Chen, Ling Deng, Zhiyong Gan, Wei-Shi Zheng, and Ruixuan Wang. 2024. FodFoM: Fake Outlier Data by Foundation Models Creates Stronger Visual Out-of-Distribution Detector. In *Proceedings of the 32nd ACM International Conference on Multimedia (MM '24), October 28-November 1, 2024, Melbourne, VIC, Australia.* ACM, New York, NY, USA, 10 pages. https://doi.org/10.1145/3664647.3681309

## 1 Introduction

Out-of-distribution (OOD) detection aims at identifying test samples that are outside the distribution of training data [25, 46, 48]. The ability to detect OOD data is crucial for AI systems, as misclassifying such OOD data as in-distribution (ID) could have severe consequences in some applications like autonomous driving, intelligent healthcare, and industrial manufacturing.

To effectively address the OOD detection problem, most existing methods [1, 16, 28, 51] rely on models trained solely on ID data. Some of them analyze potential differences between ID and OOD samples by examining outputs at various locations in the model, such as the softmax output layer, the penultimate or other intermediate layer. Any observed difference could be employed to help design a more effective OOD detection score function [26, 53, 64, 69]. Some other methods like SSD+ [47] and CIDER [35] demonstrate that encouraging the model to learn more compact ID representations in the representation space improves OOD detection performance. However, models with these OOD detection methods still exhibit overconfidence in predicting OOD data as ID class(es) [59] and often struggle with effective OOD detection due to the lack of OOD information during model training. To better address this issue,

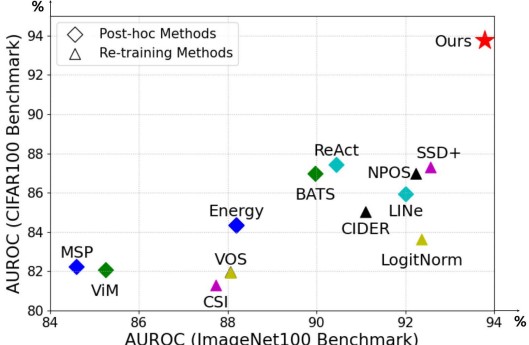

**Figure 1: OOD detection performance of different methods on the CIFAR100 and ImageNet100 benchmarks (see Section 3.1 for benchmark details). ◇: post-hoc approaches. △: training-based approaches. Ours belongs to the latter.**

some methods like outlier exposure (OE) [17] utilize auxiliary OOD samples to train the model and improve decision boundaries between ID and OOD samples. Considering the difficulty in obtaining real auxiliary OOD samples, various generative methods have been proposed to generate fake OOD data, e.g., synthesizing fake OOD images by adversarial generative networks (GANs) [23, 36] or constructing fake OOD features in the feature space as in VOS [12] and NPOS [55]. However, it is challenging for GANs to stably generate even simple images, and the generalizability of VOS and NPOS is limited by the simplified assumption in feature distribution and the complex sampling process. Although the recently proposed Dream-OOD[11] can learn to generate realistic and semantically rich OOD images based on a diffusion model, the only usage of class names to control the generation process affects the scalability.

Recently, foundation models [4] with extensive prior knowledge and strong generalization capabilities have been applied in various domains. For example, BLIP-2 [27] possesses powerful image-text generation ability [19, 50] by combining vision and language models. CLIP [41], which holds strong text-image alignment ability, has been adopted by many large-scale models [29, 42]. Stable Diffusion [44] is known for its ability to use conditional controls to generate photo-realistic images. Besides, being capable of detecting everything, GroundingDINO [30] has been used in open-world.

To fully leverage the robust capabilities of the foundation models and address the limitations of current fake OOD data generation methods, we propose a novel framework called FodFoM (**F**ake **o**utlier **d**ata by **Fo**undation **M**odels) for model training. Our framework generates two types of fake OOD data to enhance OOD detection performance. The first type is generated by combining BLIP-2, CLIP, and Stable Diffusion. Specifically, we utilize BLIP-2 and class names to generate exhaustive descriptions for training images of each ID class. These descriptions are then encoded into the textual space using CLIP's text encoder. Based on these text embeddings of ID classes and corresponding mean text embedding of each class, fake OOD text embeddings which are similar to the ID text embeddings are constructed, and such fake OOD text embeddings are then utilized as conditions to guide the Stable Diffusion model to generate challenging fake OOD images that closely resemble the ID

classes. The second type is generated using GroundingDINO. In particular, GroundingDINO's capability to "detect everything" is leveraged to identify ID objects in training images. After blurring the detected ID object regions in ID images, manipulated background images without clear ID objects can be obtained. Such manipulated images are highly correlated with the original ID images and used as the second type of challenging fake OOD images. By training the model with these two types of challenging fake OOD images along with ID images, the model can learn to better differentiate OOD images from ID images. Extensive experiments demonstrate that our proposed method achieves superior OOD detection performance across a wide range of benchmarks. The contributions of this study are summarized below.

- A novel framework FodFoM is proposed which utilizes the collaboration of BLIP-2, CLIP, Stable Diffusion, and GroundingDINO to automatically generate fake OOD images. These fake OOD images can help enhance the model's ability to effectively detect OOD images.
- The proposed framework introduces a simple effective method for generating fake text representations by leveraging CLIP's latent text space. By using these representations as conditions, Stable Diffusion can generate OOD samples that are semantically similar to the ID samples.
- Extensive validations on multiple OOD detection benchmarks were performed, with state-of-the-art performance achieved by our method.

## 2 Method

In the OOD detection task, the model is expected to identify whether or not a new data belongs to one of the learned ID classes. OOD detection can be viewed as a binary classification task. A scoring function $S_\lambda$ is usually designed based on the model's output at a particular layer for OOD detection, where $\lambda$ is a threshold. In the testing phase, any new sample resulting in a larger score than $\lambda$ is determined as ID, otherwise it is determined as OOD.

### 2.1 Overview

Our framework FodFoM generates fake OOD images in two ways (Figure 2). ① The first way employs three foundation models, i.e., BLIP-2, CLIP, and Stable Diffusion, to innovatively help construct challenging fake OOD images. Basically, textual description of each ID image is first generated by the image-to-text model BLIP-2, and the combination of the textual description and a simple class-specific prompt is then sent to the CLIP's text encoder to obtain the text embedding for the ID image. Based on text embeddings of all images for each ID class, challenging fake OOD text embeddings outside the cluster of ID text embeddings are innovatively constructed and then used as conditions to control Stable Diffusion to generate challenging fake OOD images. These fake OOD images are similar to but different from ID images, therefore can help the classifier find a more compact decision boundary around each ID class to improve the OOD detection ability. ② The second way employs the foundation model GroundingDINO to help construct background images as fake OOD images. The foreground object in each ID image is first detected by GroundingDINO, and then the foreground region is blurred to create a background image

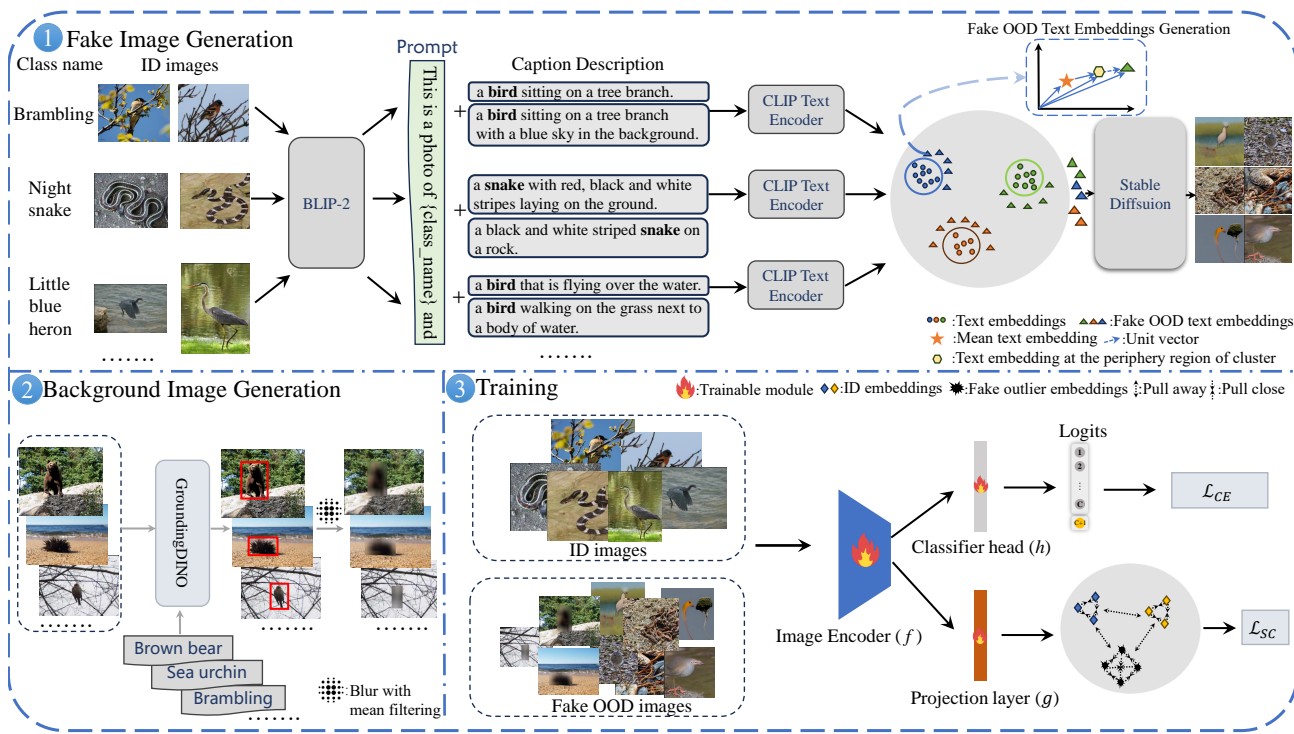

**Figure 2: Illustration of the proposed FodFoM framework. FodFoM generates fake OOD images in two ways. ①: Text description of each image is generated by BLIP-2 and then the slightly augmented description with a class-specific prompt is mapped to the textual semantic space by CLIP's text encoder. The fake OOD text embeddings are then constructed using the proposed method and finally sent to Stable Diffusion to generate challenging fake OOD images. ②: The semantic regions of ID objects (detected by GroundingDINO) are blurred to obtain background images as fake OOD images. During training (③), fake OOD images are labeled with an additional class different from ID classes.**

for the ID image. Considering that OOD images sharing similar background regions with ID images are often misclassified as ID classes [22, 43], such background images as fake OOD images will help train a classifier which can better focus on foreground region during inference. Together with training images of ID classes, these two types of fake OOD images can help train a classifier model having better OOD detection performance.

## 2.2 Fake Image Generation

The first way to generate fake OOD images is based on foundation models BLIP-2, CLIP, and Stable Diffusion. This generation method is motivated by Chen et al. [6] and Ming et al. [34]'s suggestion that more challenging OOD data help the model find a better decision boundary between ID and OOD classes, as well as by the question "How to generate effective fake OOD images using as much ID semantic information as possible?". Built on BLIP-2's powerful image-to-text capability and Stable Diffusion's strong image-generation ability with text as conditions, the proposed method can generate fake OOD images whose semantic contents are similar to but different from ID images. It consists of three components which are detailed below.

*2.2.1 Caption generation.* An initial caption description is generated by BLIP-2 for each ID image, denoted by $\text{BLIP2}(\mathbf{x}_i)$ for the $i$-th

training ID image $\mathbf{x}_i$. Note that although BLIP-2 can generate rich and precise content descriptions for ID images, it does not discriminate the names of ID classes at a fine-grained level. For example, in the upper left part of Figure 2, BLIP-2 can recognize *bird* but the output caption is not fine-grained to *Brambling* or *Little blue heron*, and can recognize *snake* but not fine-grained to *Night snake*. To fine-grain the ID class information in the description to guide Stable Diffusion in generating ID-similar OOD images in the next stage, a simple class-specific prompt $P(\mathbf{x}_i)$, i.e., "This is a photo of $class\_name(\mathbf{x}_i)$ and", is prefixed to the initial caption description for each image, where $class\_name(\mathbf{x}_i)$ denotes the name of the image class for image $\mathbf{x}_i$. The resulting final textual description $T_i$ for image $\mathbf{x}_i$ is

$$T_i = [P(\mathbf{x}_i), \text{BLIP2}(\mathbf{x}_i)], \tag{1}$$

where $[a, b]$ denotes the concatenation of two sentences $a$ and $b$.

*2.2.2 Generation of fake OOD text embeddings.* To guide Stable Diffusion to generate effective fake OOD images that look like ID images, some textual descriptions similar to the ID description are needed. Considering that the textual conditions accepted by Stable Diffusion are vectorized and based on the textual semantic space of CLIP, the text embedding for $T_i$ is generated by mapping the description $T_i$ into CLIP's textual semantic space via CLIP's text

encoder $\mathcal{T}$ as follows,

$$\mathbf{t}_i = \mathcal{T}(T_i) . \tag{2}$$

With the powerful textual representation ability from the CLIP's

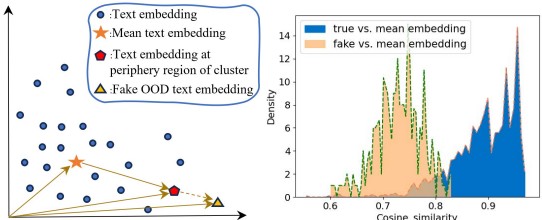

**Figure 3: Generation of fake OOD text embeddings. Left: Construction of fake OOD embeddings in the demonstrative textual semantic space. Right: Distribution (blue) of cosine similarities between the mean text embedding of the class "Speedboat" and text embeddings ("true") of ID images from this class in ImageNet100, and the distribution (yellow) between the mean text embedding and the challenging fake OOD embeddings ("fake") for this class.**

text encoder and the desired text-image alignment ability of CLIP, the text embedddings of ID images are often clustered for each ID class and separated between ID classes in the textual semantic space. For each ID class, any new text embedding which is close to but outside the embedding cluster of the ID class does not belong to this ID class and meanwhile will probably be away from any other ID class. Such new embeddings will be generated as fake OOD text embeddings, with the generation process detailed below.

For each ID class, the first step is to identify the text embeddings of the ID images which are located at the periphery region of the associated embedding cluster. Formally, for the $c$-th ID class, let $\boldsymbol{\mu}_c$ denote the mean of text embeddings over all the ID images of this class, and $\mathcal{J}_c$ denote the index set of all selected text embeddings from the $c$-th class such that for any $j \in \mathcal{J}_c$, the cosine similarity (also refer to Supplementary Section 4 for other similarity measurements) between the text embedding $\mathbf{t}_j$ and the mean embedding $\boldsymbol{\mu}_c$ is not higher than a threshold $s_\alpha$. $s_\alpha$ is automatically determined such that the size of the index set $\mathcal{J}_c$ accounts for $\alpha\%$ of the total ID images of the $c$-th class, where $\alpha$ is a hyperparameter. The selected text embeddings associated with the index set $\mathcal{J}_c$ are relatively less similar to the mean embedding $\boldsymbol{\mu}_c$ and therefore located at the periphery region of the embedding cluster for the $c$-th class.

Then, for each text embedding $\mathbf{t}_j$ associated the index $j \in \mathcal{J}_c$, a new text embedding $\mathbf{t}_j^o$ close to $\mathbf{t}_j$ but slightly outside the cluster of the $c$-th class can be obtained by

$$\mathbf{t}_j^o = \mathbf{t}_j + \gamma \cdot \boldsymbol{\omega}_j , \tag{3}$$

where $\boldsymbol{\omega}_j = (\mathbf{t}_j - \boldsymbol{\mu}_c)/||\mathbf{t}_j - \boldsymbol{\mu}_c||$ is the unit vector from the mean embedding $\boldsymbol{\mu}_c$ to the text embedding $\mathbf{t}_j$, and $\gamma$ is the step length hyperparameter. The constructed text embeddings $\{\mathbf{t}_j^o\}$ for each ID class are collected together to form the set of fake OOD text embeddings. As Figure 3 (Right) demonstrates, similarities between constructed fake OOD text embeddings and the mean embedding are smaller than those for most true ID text embeddings as expected.

*2.2.3 Fake OOD Image Generation by Stable Diffusion.* The constructed fake OOD text embeddings are used as conditions to guide Stable Diffusion to generate challenging fake OOD images. Stable Diffusion is able to control image generation using data from other modalities as conditions [38, 45]. For text modality, text embeddings from the CLIP's text encoder are employed as conditions. Therefore, the constructed fake OOD text embeddings in the textual semantic space of the CLIP's text encoder can be directly used as conditions to control Stable Diffusion to generate images whose visual contents are consistent with the semantic information of the fake OOD text embeddings. Since fake OOD text embeddings are semantically similar to but different from text embeddings of ID images, the generated images from Stable Diffusion with the fake OOD text embeddings as conditions would also be similar to but different from ID images, therefore can be used as challenging fake OOD images.

## 2.3 Background Image Generation

The other way to generate fake OOD images is based on GroundingDINO which has impressive capability in detecting open set objects by leveraging additional texts as queries. This method is inspired by Kim et al. [22] and Ren et al. [43]'s discovery that OOD images sharing similar background with ID images often mislead the model into thinking that these OOD images are semantically similar to ID images and thus incorrectly predicting such OOD images as ID classes with high confidence.

To allow the classifier model to focus more on the foreground objects during model training while being aware of the background information, pure background images are constructed as fake OOD images based on ID images and powerful object detection capability of GroundingDINO (Figure 2, lower left). Specifically, each ID training image along with the textual class name of the image is fed to GroundingDINO to obtain the bounding box containing the foreground object of the ID class. Then, the image region within the bounding box is blurred by a mean filtering operator, resulting in a background image without clear ID object information. For those ID images whose foreground objects occupy the majority of the whole image region, background images would contain much less background information after blurring foreground regions. Therefore, only when area of the foreground region is smaller than a predefined proportion $\beta\%$ (e.g., 50%) of the whole ID image, the ID image is used to construct a background image as described above.

## 2.4 Model Training

The fake OOD images generated by the above two methods are used together with the ID images to train the classifier's image encoder $f$, the classifier head $h$, and the projection layer $g$ (Figure 2, lower right). In order for the classifier to distinguish between the challenging OOD images and images of various ID classes, the fake OOD images are considered as an additional class. If there are totally $C$ ID classes, then the output of the classifier is $(C + 1)$-dimensional, and the commonly used cross-entropy loss $\mathcal{L}_{CE}$ is adopted to train the image encoder $f$ and the classifier head $h$.

In addition, to further separate the individual ID classes and the the class of fake OOD images in the feature space, the supervised contrastive loss $\mathcal{L}_{SC}$ is adopted to train the model following the

**Table 1: OOD detection performance on the CIFAR10 and CIFAR100 benchmarks with model backbone ResNet18. ↑ indicates that larger values are better and ↓ indicates that smaller values are better. The best and second-best results are indicated in bold and underline. All values are percentages.**

| ID Dataset Model | Method | OOD Datasets | | | | | | | | | | | | Average | |
| --- | --- | --- | --- | --- | --- | --- | --- | --- | --- | --- | --- | --- | --- | --- | --- |
| | | SVHN | | LSUN-R | | LSUN-C | | iSUN | | Textures | | Places365 | | | |
| | | FPR95↓ | AUROC↑ | FPR95↓ | AUROC↑ | FPR95↓ | AUROC↑ | FPR95↓ | AUROC↑ | FPR95↓ | AUROC↑ | FPR95↓ | AUROC↑ | FPR95↓ | AUROC↑ |
| CIFAR10 ResNet18 | MSP | 61.22 | 86.99 | 41.62 | 93.84 | 34.30 | 95.40 | 43.14 | 93.21 | 53.40 | 90.19 | 54.51 | 88.74 | 48.03 | 91.40 |
| | Mahalanobis | 67.25 | 89.51 | 48.37 | 92.38 | 91.65 | 74.55 | 44.24 | 92.68 | 45.92 | 91.96 | 66.11 | 85.79 | 60.59 | 87.81 |
| | ODIN | 53.56 | 77.48 | 17.31 | 94.63 | 13.64 | 96.09 | 19.87 | 93.55 | 46.65 | 80.85 | 49.72 | 79.92 | 33.46 | 87.09 |
| | Energy | 41.25 | 87.69 | 24.19 | 95.01 | 11.37 | 97.63 | 26.40 | 94.16 | 42.52 | 89.10 | 40.04 | 88.71 | 30.96 | 92.05 |
| | ViM | 53.75 | 88.67 | 34.17 | 94.34 | 82.31 | 87.18 | 31.41 | 94.25 | 36.15 | 92.83 | 49.64 | 88.86 | 47.90 | 91.02 |
| | DICE | 36.42 | 91.46 | 31.57 | 93.77 | 7.10 | 98.67 | 36.94 | 92.05 | 47.02 | 88.41 | 46.74 | 86.05 | 34.30 | 91.73 |
| | BATS | 41.42 | 87.84 | 24.17 | 95.02 | 11.35 | 97.63 | 26.36 | 94.16 | 42.13 | 89.29 | 40.04 | 88.71 | 30.91 | 92.11 |
| | ReAct | 43.19 | 87.56 | 24.82 | 95.12 | 12.23 | 97.53 | 26.90 | 94.31 | 41.95 | 90.02 | 40.78 | 89.00 | 31.65 | 92.26 |
| | DICE+ReAct | 36.90 | 91.31 | 31.59 | 93.71 | 7.29 | 98.64 | 37.15 | 92.10 | 46.76 | 88.61 | 46.76 | 86.12 | 34.41 | 91.75 |
| | FeatureNorm | **2.37** | **99.45** | 33.42 | 94.71 | **0.10** | **99.93** | 27.01 | 95.65 | 23.03 | 95.65 | 58.96 | 87.95 | 24.14 | 95.55 |
| | LINe | 45.38 | 87.96 | 39.25 | 92.61 | 9.75 | 98.19 | 41.52 | 91.74 | 58.37 | 84.14 | 53.02 | 85.70 | 41.22 | 90.06 |
| | CSI | 37.38 | 94.69 | 13.13 | 97.51 | 10.63 | 97.93 | 10.36 | 98.01 | 28.85 | 94.87 | 38.31 | 93.04 | 23.11 | 96.01 |
| | SSD+ | 2.47 | 99.51 | 46.72 | 93.89 | 10.56 | 97.83 | 28.44 | 95.67 | 9.27 | 98.35 | 22.05 | 95.57 | 19.92 | 96.80 |
| | VOS | 35.73 | 93.74 | 25.54 | 95.29 | 18.47 | 96.55 | 30.17 | 94.16 | 44.16 | 90.07 | 44.18 | 88.13 | 33.04 | 92.99 |
| | LogitNorm | 12.68 | 97.75 | 15.29 | 97.45 | 0.53 | 99.82 | 15.36 | 97.43 | 31.56 | 94.09 | 32.31 | 93.92 | 17.96 | 96.75 |
| | NPOS | 8.49 | 96.93 | 19.44 | 95.69 | 4.26 | 98.38 | 20.37 | 95.21 | 31.04 | 94.15 | 40.13 | 90.89 | 20.62 | 95.21 |
| | CIDER | 2.89 | **99.72** | 23.13 | 96.28 | 5.45 | 99.01 | 20.21 | 96.64 | 12.33 | 96.85 | 23.88 | 94.09 | 14.64 | 97.10 |
| | **FodFoM (Ours)** | 7.22 | 98.54 | **9.17** | **98.23** | 9.35 | 98.12 | **8.02** | **98.45** | **6.15** | **98.78** | **10.69** | **97.87** | **8.43** | **98.33** |
| CIFAR100 ResNet18 | MSP | 69.74 | 84.73 | 66.89 | 85.65 | 77.08 | 81.83 | 69.40 | 84.77 | 80.08 | 77.65 | 78.38 | 78.81 | 73.60 | 82.24 |
| | Mahalanobis | 92.62 | 66.80 | 89.00 | 68.46 | 98.83 | 49.58 | 88.45 | 68.44 | 72.68 | 74.57 | 92.87 | 63.26 | 89.07 | 65.18 |
| | ODIN | 79.74 | 81.40 | 37.63 | 93.21 | 72.66 | 85.93 | 39.59 | 92.58 | 73.07 | 80.42 | 80.39 | 77.22 | 63.85 | 85.13 |
| | Energy | 68.90 | 87.66 | 59.71 | 88.58 | 73.21 | 84.46 | 64.03 | 87.50 | 79.61 | 78.22 | 77.74 | 79.64 | 70.53 | 84.34 |
| | ViM | 73.70 | 84.45 | 61.30 | 88.05 | 92.76 | 69.87 | 61.92 | 87.34 | 57.93 | 86.31 | 81.01 | 76.54 | 71.43 | 82.09 |
| | DICE | 53.60 | 90.22 | 79.84 | 81.17 | 40.03 | 92.52 | 79.79 | 80.96 | 78.65 | 77.46 | 82.31 | 76.76 | 69.04 | 83.18 |
| | BATS | 62.05 | 89.31 | 50.38 | 91.21 | 73.70 | 84.55 | 55.97 | 90.30 | 72.93 | 84.50 | 72.61 | 82.03 | 64.61 | 86.98 |
| | ReAct | 58.24 | 90.02 | 50.82 | 90.98 | 70.70 | 85.75 | 55.91 | 90.18 | 70.85 | 85.39 | 71.85 | 82.25 | 63.06 | 87.43 |
| | DICE+ReAct | 48.20 | 91.19 | 84.18 | 78.79 | 32.05 | 93.71 | 82.23 | 79.65 | 66.74 | 83.96 | 80.28 | 77.96 | 65.61 | 84.21 |
| | FeatureNorm | **15.98** | 96.59 | 96.57 | 61.80 | **4.56** | **98.95** | 93.56 | 65.15 | 51.67 | 83.54 | 93.61 | 56.83 | 59.33 | 77.07 |
| | LINe | 52.02 | 91.01 | 65.66 | 86.87 | 47.76 | 91.23 | 69.27 | 85.90 | 71.22 | 83.37 | 80.90 | 77.21 | 64.47 | 85.93 |
| | CSI | 54.62 | 91.28 | 87.73 | 76.63 | 81.53 | 80.18 | 83.88 | 77.70 | 76.45 | 84.25 | 84.27 | 77.81 | 78.08 | 81.31 |
| | SSD+ | 16.04 | **97.01** | 79.35 | 84.55 | 56.51 | 91.16 | 78.38 | 84.53 | 54.75 | 89.42 | 78.17 | 77.17 | 60.53 | 87.30 |
| | VOS | 78.36 | 80.58 | 69.77 | 84.77 | 77.38 | 83.61 | 69.65 | 85.48 | 76.60 | 80.48 | 80.47 | 77.57 | 75.37 | 81.96 |
| | LogitNorm | 51.34 | 91.79 | 88.80 | 78.67 | 6.82 | 98.70 | 90.16 | 75.55 | 77.02 | 77.52 | 77.79 | 79.56 | 65.32 | 83.63 |
| | NPOS | 32.58 | 92.17 | 49.72 | 89.45 | 39.26 | 91.82 | 65.27 | 86.57 | 62.93 | 84.21 | 65.48 | 77.63 | 52.54 | 86.98 |
| | CIDER | 31.36 | 93.47 | 80.39 | 81.54 | 43.68 | 89.45 | 78.23 | 81.33 | 35.51 | 91.70 | 82.80 | 72.71 | 58.66 | 85.03 |
| | **FodFoM (Ours)** | 33.19 | 94.02 | **28.24** | **95.09** | 26.79 | 95.04 | **33.06** | **94.45** | 35.44 | 93.38 | **42.30** | **90.68** | **33.17** | **93.78** |

idea of SupCon [21] as follows,

$$\mathcal{L}_{SC} = -\frac{1}{N}\sum_{i=1}^{N}\frac{1}{|P(i)|}\sum_{p\in P(i)}\log\frac{\exp\left(s(\mathbf{z}_i,\mathbf{z}_p)/\tau\right)}{\sum_{a\in A(i)}\exp\left(s(\mathbf{z}_i,\mathbf{z}_a)/\tau\right)}, \quad (4)$$

where $N$ is total number of training ID images and fake OOD images, $A(i)$ represents all the sample indices in the mini-batch that includes the sample with index $i$, and $P(i)$ is the subset of $A(i)$ in which all the corresponding samples share the same class label as that of the sample with index $i$. $\mathbf{z} = g(f(\mathbf{x}))$ is the projected visual embedding for the input image $\mathbf{x}$, where the structure of the projection layer $g$ is *Linear-BatchNorm-ReLU-Linear* and $s(\mathbf{z}_i,\mathbf{z}_p)$ represents the cosine similarity between the two embeddings $\mathbf{z}_i$ and $\mathbf{z}_p$. $\tau$ is the temperature scaling factor.

The two loss functions $\mathcal{L}_{CE}$ and $\mathcal{L}_{SC}$ help separate ID images from OOD images at two different levels, and the combination of them is utilized to optimize the classifier model, i.e., by minimizing the joint loss function $\mathcal{L}$,

$$\mathcal{L} = \mathcal{L}_{CE} + \lambda\mathcal{L}_{SC}, \quad (5)$$

where coefficient $\lambda$ is used to balance the two loss terms.

## 2.5 Model Inference

Once the model is well-trained, the image encoder and the classifier head are used to determine whether any new image is ID or OOD. Here the Energy-based OOD detection score from the post-hoc method ReAct [51] is adopted, although other post-hoc methods can also be used. The Energy is defined as

$$E(\mathbf{x}) = -\log\sum_{i=1}^{C}\exp(h_i(f(\mathbf{x}))), \quad (6)$$

where $h_i(f(\mathbf{x}))$ represents the $i$-th logit generated for image $\mathbf{x}$ after passing through the image encoder $f$ and the classifier head $h$. The Energy score [31] is defined as the negative Energy, i.e., $-E(\mathbf{x})$, with larger score indicating the image is more likely from certain ID class. The OOD detection score from ReAct achieves a better performance by pruning high-activation values of the features from the penultimate layer before calculating the Energy score. Note that only the logit values of the $C$ ID classes are used to calculate the OOD detection score, and the potential utilization of the logit associated with the OOD class has also been discussed in Supplementary Section 3.

# 3 Experiments

## 3.1 Experimental Setup

Our method is extensively evaluated on three OOD detection benchmarks, including two small-scale CIFAR benchmarks [24] and one large-scale ImageNet benchmark [55]. Each benchmark contains one ID training set, one ID test set and multiple OOD test sets. For the two CIFAR benchmarks, CIFAR10 and CIFAR100 are respectively used as ID dataset, and SVHN [37], LSUN-R [62], LSUN-C [62], iSUN [61], Textures [8], Places365 [68] are used as OOD test sets for performance evaluation. For the ImageNet benchmark, ImageNet100 is used as the ID dataset, and four test OOD datasets, iNaturalist [56], SUN [60], Places [68], and Textures [8], are used for evaluation. The categories of OOD datasets are disjoint from those of the ID dataset. Please see Supplementary Section 1 for more dataset details.

We follow the common practice as previous studies [35, 51, 55] to use ResNet18 or ResNet34 [14] as the model backbone on the CIFAR10 and CIFAR100 benchmarks, and ResNet50 or ResNet101 on the ImageNet100 benchmark. For generation of the first type of fake OOD images, BLIP-2 leveraging OPT-2.7b [67], CLIP-L/14 based on ViT-L/14 as CLIP's Text Encoder, and Stable Diffusion v1.4 are used. For generation of the second type of fake OOD images, GroundingDINO with the Swin-T [32] backbone is used. The hyperparameter $\alpha$ is set to 30 for CIFAR10 and CIFAR100 benchmarks, and 20 for ImageNet100 benchmark. The step size $\gamma$ is respectively set to 1e-5, 5e-5 and 9e-5 for ImageNet100, such that each selected periphery text emebdding generates three fake OOD text embeddings. For CIFAR benchmarks, $\gamma$ is respectively set to 3e-5, 6e-5, 9e-5, 1.2e-4, 1.5e-4, such that each selected text embedding generates five fake OOD text embeddings. The Mean filtering function with kernel size 50 is used to blur foreground regions during background image generation. The proportion hyperparameter $\beta\%$ is set to 50%. The dimension of the hidden layer in the projection layer is the same as that of input to the projection layer, and the output layer of the projection layer is 128-dimensional.

During model training, each ID image or fake OOD image was randomly cropped and resized to $32 \times 32$ pixels for the CIFAR datasets or $224 \times 224$ pixels for the ImageNet100 training set, while maintaining the aspect ratio within a scale range of 0.2 to 1. Random horizontal flipping, color jittering, and grayscale transformation were applied to each image. The stochastic gradient descent optimizer with momentum 0.9 and weight decay 1e-4 was used to train each model up to 200 epochs on three ID training datasets. The initial learning rate was 0.05, with warming up from 0.01 to 0.05 in the first 10 epochs when the batch size was larger than 256, and it was decayed by a factor of 10 at the 100-th and 150-th epoch on both CIFAR benchmarks and at the 100-th, 150-th, 180-th epoch on ImageNet100 benchmark. The batch size was set to 512 for CIFAR benchmarks and 128 for ImageNet100 benchmark. The temperature $\tau$ was set to 0.1 and coefficient $\lambda$ was set to 1 for all experiments. During testing, only random cropping and resizing were performed on each test image. All experiments were run on NVIDIA GeForce A30 GPUs.

The two widely adopted metrics FPR95 and AUROC were used for performance evaluation. FPR95 represents the false positive rate of OOD test samples when the true positive rate of ID test samples

reaches 95%, and AUROC is the area under the receiver operating characteristic curve. The lower FPR95 and higher AUROC indicate better OOD detection performance. The performance on each OOD test set with respect to the corresponding ID test set and the average performance over all OOD test sets were reported.

**Table 2: OOD detection performance on the CIFAR100 benchmark with ResNet34 backbone and the ImageNet100 benchmark with ResNet50 and ResNet101 backbones. Averaged performances over six OOD datasets for the CIFAR benchmark and four OOD datasets for the ImageNet100 benchmark were reported.**

| Method | CIFAR-100 ResNet34 | | ImageNet100 ResNet50 | | ImageNet100 ResNet101 | |
|---|---|---|---|---|---|---|
| | FPR95↓ | AUROC↑ | FPR95↓ | AUROC↑ | FPR95↓ | AUROC↑ |
| MSP | 78.29 | 79.25 | 68.28 | 84.60 | 60.43 | 87.30 |
| Mahalanobis | 93.86 | 55.21 | 82.70 | 54.88 | 78.83 | 66.47 |
| ODIN | 64.01 | 83.44 | 49.83 | 89.78 | 59.67 | 86.62 |
| Energy | 69.41 | 83.64 | 59.85 | 88.20 | 51.98 | 90.33 |
| ViM | 61.51 | 85.00 | 67.39 | 85.25 | 61.15 | 86.51 |
| DICE | 68.14 | 83.53 | 36.46 | 92.11 | 37.02 | 92.65 |
| BATS | 56.73 | 87.52 | 49.60 | 89.97 | 55.02 | 88.02 |
| ReAct | 50.56 | 88.30 | 44.06 | 90.45 | 48.67 | 90.35 |
| DICE+ReAct | 65.61 | 84.22 | 34.75 | 92.39 | 36.72 | 91.67 |
| FeatureNorm | 59.60 | 80.12 | 60.39 | 83.80 | 60.37 | 83.64 |
| LINe | 54.52 | 86.32 | 36.19 | 92.01 | 54.50 | 89.52 |
| CSI | 70.59 | 83.58 | 57.52 | 87.73 | 56.08 | 86.48 |
| SSD+ | 57.98 | 88.26 | 39.50 | 92.56 | 37.70 | 92.83 |
| VOS | 80.53 | 79.03 | 57.85 | 88.07 | 56.99 | 88.65 |
| LogitNorm | 70.91 | 79.44 | 38.64 | 92.37 | 36.21 | 92.94 |
| NPOS | 45.84 | 88.31 | 36.13 | 92.23 | 35.97 | 92.50 |
| CIDER | 50.66 | 86.70 | 48.03 | 91.11 | 50.31 | 90.12 |
| **FodFoM (Ours)** | **44.62** | **91.34** | **33.44** | **93.79** | **34.18** | **93.73** |

## 3.2 Efficacy Evaluation of the Method

*3.2.1 Evaluation on Common Benchmarks.* Table 1 summarizes the performance comparison of our method with various strong baselines on the CIFAR10 and CIFAR100 benchmarks. These baselines contain multiple post-hoc methods, which do not require model retraining, including MSP [16], Mahalanobis [26], ODIN [28], DICE [52], ViM [58], Energy [31], BATS [69], DICE+ReAct [52], ReAct [51], FeatureNorm [63] and LINe [1]. In addition, since our method is based on model training, our method is also compared with various powerful re-training methods, including CSI [54], SSD+ [47], VOS [12], LogitNorm [59], NPOS [55] and CIDER [35]. As Table 1 shows, our method achieves state-of-the-art performance on four of the six OOD datasets and superior average performance on both benchmarks. For example, on the CIFAR10 benchmark with ResNet18 (Table 1, upper half), our method's average performance outperforms the best baseline CIDER by a large margin (FPR95 **8.43%** vs. 14.64%, AUROC **98.33%** vs. 97.10%).

Similar results were observed on the CIFAR100 benchmark with ResNet18 (Table 1, lower half), with ResNet34 backbone (Table 2, left; also see detailed performance on CIFAR benchmarks with ResNet34 on six OOD datasets in Supplementary Table 1), and on ImageNet100 benchmark with both ResNet50 and ResNet101 backbones (Table 2, center and right; also see detailed performance on each OOD dataset in Supplementary Table 2). All results supports that the classifiers trained with the constructed fake OOD images learn better decision boundaries for effective OOD detection.

**Table 3: Evaluation on clean OOD datasets (NINCO, OpenImage-O). All values are percentages. Model backbone is ResNet50.**

| Dataset | Metrics | Methods | | | | | |
|---|---|---|---|---|---|---|---|
| | | LINe | SSD+ | VOS | LogitNorm | CIDER | FodFoM (Ours) |
| NINCO | FPR95↓ | 73.12 | 61.76 | 86.90 | 63.97 | 70.72 | **57.98** |
| | AUROC↑ | 76.25 | 85.99 | 74.57 | 85.75 | 83.06 | **87.31** |
| OpenImage-O | FPR95↓ | 62.15 | 49.60 | 86.30 | 63.24 | 63.53 | **48.68** |
| | AUROC↑ | 83.09 | 90.95 | 76.21 | 86.50 | 86.65 | **91.87** |

**Table 4: Ablation study of the proposed framework.**

| $\mathcal{L}_{SC}$ | BG-OOD | SD-OOD | CIFAR100 ResNet-18 | | ImageNet100 ResNet-50 | |
|---|---|---|---|---|---|---|
| | | | Average | | Average | |
| | | | FPR95↓ | AUROC↑ | FPR95↓ | AUROC↑ |
| | | | 63.06 | 87.43 | 44.06 | 90.45 |
| ✔ | | | 58.08 | 88.02 | 40.54 | 92.68 |
| | ✔ | | 53.32 | 88.65 | 35.49 | 92.94 |
| | | ✔ | 40.85 | 91.79 | 36.32 | 93.03 |
| | ✔ | ✔ | 35.31 | 93.17 | 35.00 | 93.12 |
| ✔ | ✔ | | 51.22 | 89.25 | 34.79 | 93.09 |
| ✔ | | ✔ | 39.56 | 92.89 | 36.22 | 93.48 |
| ✔ | ✔ | ✔ | **33.17** | **93.78** | **33.44** | **93.79** |

*3.2.2 Evaluation on Clean and Challenging OOD Benchmarks.* As one recent study [3] found, most of the currently used test OOD datasets have an unclean issue, i.e., the OOD test datasets actually contain more or less images of objects belonging to ID classes in ImageNet-1k [9]. To further validate the effectiveness of our method, we used ImageNet100 as the ID dataset and two clean and challenging OOD test sets NINCO [3] and OpenImage-O [58]. As shown in Table 3, our method again outperforms these competitors on both OOD test sets, further confirming the effectiveness of constructed fake OOD images for OOD detection.

## 3.3 Ablation Study

Ablation analysis was performed to validate the effectiveness of key components in the proposed framework for OOD detection. As Table 4 shows, the addition of $\mathcal{L}_{SC}$ (row 2) improves the performance of the original baseline (row 1), indicating that optimization in the feature space is helpful for OOD detection. The introduction of background images generated by GroundingDINO ('BG-OOD', row 3) or ID-similar fake OOD images generated by Stabel Diffusion (with BLIP-2 and CLIP) ('SD-OOD', row 4) outperforms the baseline without any components (row 1) or with $\mathcal{L}_{SC}$ (row 2), resulting in the current state-of-the-art OOD detection performance in AUROC. The performance is further improved when both types of fake OOD images were involved (row 5). When either type of fake OOD images is combined with $\mathcal{L}_{SC}$ (row 6 or row 7), the model performs better than that without $\mathcal{L}_{SC}$ (row 3 or row 4), further confirming the effectiveness of $\mathcal{L}_{SC}$. Inclusion of all fake OOD images and $\mathcal{L}_{SC}$ (last row) achieves the new state-of-the-art performance, supporting that constructed fake OOD images and supervised contrastive learning in the feature space are helpful for learning better decision boundaries for OOD detection.

## 3.4 Sensitivity and Flexibility Studies

Our proposed framework is insensitive to the value choice of hyperparameters, including the temperature factor $\tau$, the loss coefficient $\lambda$, and the threshold $\alpha$ for selecting periphery text embeddings. As shown in Figure 4, on both CIFAR10 and CIFAR100 benchmarks the performance of our method is stable and better than best baseline (CIDER both in AUROC and FPR95 for CIFAR10; ReAct in AUROC, NPOS in FPR95 for CIFAR100) for moderate adjustments of $\tau$ in the range [0.01, 4], $\lambda$ in the range [0.05, 2] and $\alpha$ in the range [10, 30].

Another merit of our method is its flexible combination with various post-hoc methods. As shown in Table 5, when applying the OOD detection scores from different post-hoc methods to our framework, each of these post-hoc methods gains a huge boost in performance compared to the original one. Please also refer to Supplementary Section 4 for flexible choice of image captioning module in our framework.

## 3.5 Visualizations of Fake OOD Images

As demonstrated in Figure 5, by combining the strengths of BLIP-2, CLIP, and Stable Diffusion with the construction of fake OOD text embeddings, fake OOD images are generated which are semantically related to ID images but do not belong to ID classes. Therefore, such generated images can serve as challenging fake OOD images to effectively improve the model's OOD detection capability. Please also refer to Supplementary Figure 2 for the constructed demonstrative background images as the other type of challenging fake OOD images.

## 4 Related Work

Multiple lines of research have been investigated for OOD detection. One line of research performs OOD detection by designing a score function based on certain observed difference in output information of the pre-trained model between ID and OOD samples, such as the softmax output [16, 28], logits output [15, 31], gradient information [7], features of the penultimate or medium layer [26, 51, 53, 58, 63, 64, 69], or sparsification operations on model weights or feature outputs [1, 10, 52]. The developed methods are also called post-hoc methods. Another line of research deals with OOD detection with additional regularization during model training. Among these methods, GODIN [18] is proposed to decompose confidence scoring during training with a modified input pre-processing method. SSD+ [47] uses supervised contrastive loss to optimize the model, proving that more compact ID representations are beneficial for OOD detection. LogitNorm [59] performs normalization on the logit of last classifier layer to mitigate overconfidence in predicting OOD as ID classes. CIDER [35] is a hyperspherical prototype learning method that allows ID data to form class prototypes during training and keeps each ID class of data close to the corresponding prototype, achieving state-of-the-art performance on the CIFAR10 benchmark. However, performance of these methods is often limited due to learning without any OOD information. In contrast, our method constructs fake OOD data similar or related to ID data to achieve superior performance.

Compared to those methods which do not use any OOD data during model training, inclusion of certain OOD training data during model training has proven to be beneficial in further improving

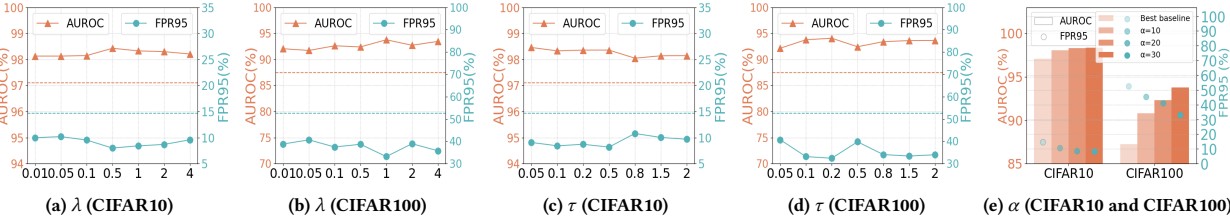

(a) $\lambda$ (CIFAR10)  (b) $\lambda$ (CIFAR100)  (c) $\tau$ (CIFAR10)  (d) $\tau$ (CIFAR100)  (e) $\alpha$ (CIFAR10 and CIFAR100)

**Figure 4: Sensitivity study of hyper-parameters $\lambda$, $\tau$, and $\alpha$. Dashed line: performance of the best baseline. Backbone: ResNet18.**

**Table 5: Fusion of our framework with various post-hoc OOD methods on three benchmarks. Paired values by '/': the left one is from the original baseline and the right one is from the fusion one. Values are average percentages over several OOD datasets.**

| Method | ResNet18 | | | | ResNet34 | | | | ImageNet100 | | | |
| | CIFAR10 | | CIFAR100 | | CIFAR10 | | CIFAR100 | | ResNet50 | | ResNet101 | |
| | FPR95↓ | AUROC↑ | FPR95↓ | AUROC↑ | FPR95↓ | AUROC↑ | FPR95↓ | AUROC↑ | FPR95↓ | AUROC↑ | FPR95↓ | AUROC↑ |
|---|---|---|---|---|---|---|---|---|---|---|---|---|
| MSP | 48.03/**13.28** | 91.40/**97.71** | 73.60/**33.43** | 82.24/**92.60** | 40.95/**12.65** | 92.09/**98.03** | 78.29/**43.15** | 79.25/**90.49** | 68.28/**44.93** | 84.60/**91.12** | 60.43/**44.95** | 87.30/**91.21** |
| Energy | 30.96/**8.91** | 92.05/**98.29** | 70.53/**35.26** | 84.34/**93.30** | 26.69/**6.69** | 93.17/**98.67** | 69.41/**48.01** | 83.64/**90.61** | 59.85/**37.72** | 88.20/**93.06** | 51.98/**39.43** | 90.33/**92.65** |
| ViM | 47.90/**8.55** | 91.02/**98.36** | 71.43/**56.22** | 82.09/**88.13** | 38.35/**4.74** | 93.75/**98.86** | 61.51/**41.15** | 85.00/**92.25** | 67.39/**64.10** | 85.25/**89.45** | 61.15/**60.67** | 86.51/**90.15** |
| ReAct | 31.65/**8.43** | 92.26/**98.33** | 63.06/**33.17** | 87.43/**93.78** | 27.76/**7.19** | 93.29/**98.57** | 50.56/**44.62** | 88.30/**91.34** | 44.06/**33.44** | 90.45/**93.79** | 48.67/**34.18** | 90.35/**93.73** |

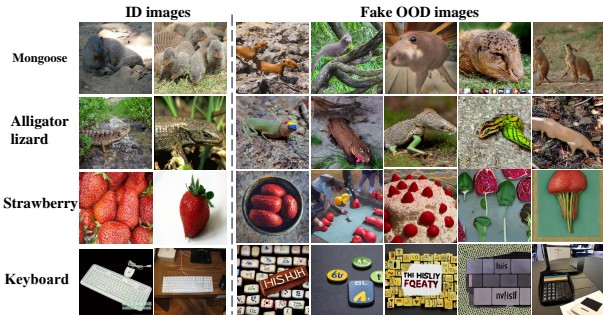

**Figure 5: Demonstrative fake OOD images generated from Stable Diffusion. ID images are from ImageNet100.**

model's OOD detection performance [20, 34]. A group of straightforward methods directly utilize additional auxiliary OOD datasets to regularize model training [17, 34]. However, collection of real OOD datasets is time-consuming and costly, making these methods ineffective when real OOD data are unavailable. In order to address this limitation, another group of methods try to construct spurious OOD data to help optimize the model, such as generating OOD images based on GANs [23, 36], constructing fake OOD images based on the image's own transformations [54, 63], and constructing pseudo-OOD features in the feature space [12, 55]. This group of methods bring OOD knowledge to the model to certain degree and enhance the model's OOD detection ability. However, these methods are often constrained, e.g., by the instability of training GANs [23, 36] or by the simplified assumption of conditional Gaussian distribution in the feature space [12, 55]. The recently proposed Dream-OOD [11] can generate fake OOD images by learning a text-conditioned latent space based on single-word ID class names and utilizing Diffusion Model in low-likelihood regions in the latent space (same as NPOS [55]), but pre-learning of the text-conditioned

space and the inability of handling multi-word class names make it challenging to generalize this method. In contrast, our method needs no such pre-learning and can construct more semantically informative descriptions which are then used to help construct fake OOD text embeddings in the latent space of CLIP's text encoder.

Recently, a variety of foundation models have been proposed, such as CLIP [41], ChatGPT [39], BLIP-2 [27], DINO [5], GroundingDINO [30], and Stable Diffusion [44]. These foundation models contain rich prior knowledge and have been applied to various tasks, such as few-shot learning [65, 66], open-set object recognition [40], and image retrieval [2] etc. In addition, OOD detection methods based on CLIP [13, 33, 49, 57] and Stable Diffusion [11] are also becoming more popular very recently. In this study, a novel framework for OOD detection is designed which can generate fake OOD images by leveraging rich knowledge and strong downstream generalization capability from multiple foundation models.

## 5 Conclusion

In this study, a novel OOD detection framework called FodFoM is proposed by jointly utilizing multiple foundation models to generate fake OOD images. Specifically, two sets of fake OOD images are constructed, with one set generated by innovatively utilizing BILP-2, CLIP, Stable Diffusion, and the other set generated by utilizing GroundingDINO. Both sets of fake OOD images are similar to but different from ID images, thus helping the classifier model find more compact decision boundaries around ID classes. Sufficient empirical evaluations confirm that the generated fake OOD images based on these foundation models are effective to improve the model's ability in OOD detection, with state-of-the-art performance achived on multiple benchmarks. Moreover, FodFoM can be easily and flexibly combined with existing post-hoc methods. We expect that this study will inspire future research in better utilizing foundation models' rich and distinct knowledge to synthesize OOD data for enhanced OOD detection.

# Acknowledgments

This work is supported in part by the National Natural Science Foundation of China (grant No. 62071502), the Major Key Project of PCL (grant No. PCL2023A09), and Guangdong Excellent Youth Team Program (grant No. 2023B1515040025).

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
