# OpenReview forum: "FodFoM: Fake Outlier Data by Foundation Models Creates Stronger Visual Out-of-Distribution Detector"
_acmmm.org/ACMMM/2024/Conference — MM2024 Poster_

### Official Review · Reviewer_gkNp · 2024-05-24

**Rating:** 4
**Confidence:** 3

**Summary:**

The paper proposes a new framework called FodFoM for Out-of-Distribution (OOD) detection. The existing methods for OOD detection either require a lot of data to train on, or they make simplifying assumptions about the data. FodFoM addresses these limitations by using a combination of different foundation models to generate fake outlier images for training. The paper shows that using these fake images can improve the accuracy of OOD detection on multiple benchmarks.

**Strengths:**

1. Novelty: The paper proposes a new approach to generate fake OOD data for training OOD detection models. This approach leverages the capabilities of multiple foundation models (BLIP-2, CLIP, Stable Diffusion, and GroundingDINO) to create more realistic and challenging fake OOD images compared to previous methods (e.g., GANs).
2. Clarity: The paper offers a clear writing and follows a logical flow.
3. Adequate evaluation: The paper mentions extensive evaluations on multiple OOD detection benchmarks and claims to achieve state-of-the-art performance.

**Limitations:**

-The generation process of fake OOD text embeddings raises concerns about potential overlap with existing ID class clusters. Given the fine-grained categories within ImageNet, how do the authors ensure these newly generated embeddings remain distinct from established clusters?

-The paper mentions leveraging GroundingDINO for background manipulation. However, it's unclear how the authors address potential recognition errors made by GroundingDINO. Are these error samples included in the training data, or is there a correction mechanism employed?

-While a subset might be used for efficiency, how do the authors ensure the chosen subset adequately reflects the fine-grained categories within ImageNet? Is there a discussion on the generalizability of the findings to the full dataset?

-The paper proposes using semantically similar text embeddings for generating OOD images. However, a potential concern exists regarding the model's generalization on in-distribution (ID) data. Did the authors investigate the impact of this approach on ID data performance? Are there any relevant experiments to support the findings?

**Suitability:**

2

---

### Official Review · Reviewer_Hfu8 · 2024-05-24

**Rating:** 3
**Confidence:** 3

**Summary:**

The paper introduces FodFoM (Fake outlier data by Foundation Models), a novel framework for improving Out-of-Distribution (OOD) detection. The core idea is to utilize multiple foundation models to generate two types of challenging fake outlier images for training classifiers. The first type of fake outlier images is created by combining BLIP-2's image captioning, CLIP's vision-language alignment, and Stable Diffusion's image generation capabilities. These images are semantically similar but distinct from in-distribution (ID) images. The second type involves using GroundingDINO's object detection to blur foreground ID objects in ID images, creating manipulated background images.

**Strengths:**

Innovative Use of Foundation Models: The framework cleverly combines the strengths of various foundation models (BLIP-2, CLIP, Stable Diffusion, GroundingDINO) to generate challenging fake OOD images, enhancing the OOD detection capability.

Improved OOD Detection Performance: Extensive experiments show that the proposed method achieves state-of-the-art performance across multiple benchmarks.

Flexibility: The framework can be combined with multiple existing OOD detection methods, providing versatility.

Automatic Fake OOD Image Generation: The approach automates the generation of fake OOD images, reducing the need for costly and time-consuming data collection.

**Limitations:**

The framework involves multiple sophisticated models, which might increase the computational complexity and the difficulty of implementation.

It seems that there are many other methods that can ease the requirement of OOD data, such as DAL [1] and [2], which I think should be included in the discussion.

The suggested method, all in all, involves the auxiliary OOD data during training. Therefore, the authors should compared with outlier exposure and its variants.


[1] Learning to Augment Distributions for Out-of-distribution Detection

[2] Contrastive Training for Improved Out-of-Distribution Detection

[2] Contrastive Training for Improved Out-of-Distribution Detection

**Suitability:**

2

---

### Official Review · Reviewer_yjkD · 2024-05-24

**Rating:** 3
**Confidence:** 3

**Summary:**

The work presents a novel framework called FodFoM for OOD detection. The method addresses the challenge of model overconfidence on OOD data by generating two types of challenging fake outlier images using a combination of multiple foundation models (BLIP-2, CLIP, Stable Diffusion, and GroundingDINO). These images are used to train classifiers to better differentiate between ID (In-Distribution) and OOD images. Extensive empirical evaluations show that this method achieves state-of-the-art OOD detection performance on multiple benchmarks.

**Strengths:**

- Adopting foundation models leads to a strong capability for OOD detection of the classifier.

- The paper includes extensive empirical evaluations on multiple benchmarks, demonstrating the superiority of the proposed method over existing baselines.

- Generation of both ID-similar and background OOD images is intriguing.

**Limitations:**

- Lack of proof in the authors' assertion "These fake OOD images are similar to but different from ID images, therefore can help the classifier find a more compact decision boundary around each ID class to improve the OOD detection ability." One must prove what has been said, it would be meaningful to observe how the decision boundary moves for each of the adopted techniques.

- Some additional clarification on the novelty of the proposed method, and how these differentiate from the context of other works that generate OOD data for training purposes would be beneficial.

- Some sections could benefit from additional clarity, particularly in explaining the integration of multiple models.

**Suitability:**

2

---

### Official Review · Reviewer_WQzB · 2024-05-25

**Rating:** 4
**Confidence:** 3

**Summary:**

The paper introduces a novel OOD detection framework that generates two types of challenging fake outlier images. The first one uses multiple foundation models: BLIP-2, CLIP, and Stable Diffusion to generate fake images, while the second one uses DinoGround to blur the object as the OOD images. These fake images are semantically similar to but distinct from in-distribution (ID) images, enhancing the model's ability to distinguish between ID and OOD data. The proposed method achieves state-of-the-art OOD detection performance across multiple benchmarks, demonstrating its effectiveness in improving classifier robustness in open-world applications.

**Strengths:**

•	The authors find a good way to combine the foundation models to create OOD images and use it to train OOD detection in classifiers,

•	The proposed method does not require any auxiliary dataset for OOD detection.

•	The paper is well-written as it is clear and easy-to-understand

•	The authors did a comprehensive experiment on CIFAR datasets compared to previous OOD detection methods.

**Limitations:**

•	In Fig.5, the authors show some qualitative results of the ood images generated with the proposed method. However, some images are ambiguous between ID and OOD. For example, the strawberries on the cake, and the keyboard on the calculator. Is there a way to guarantee the OOD images are truly OOD? Is there a quantitative way to measure such?

•	The foundation models are usually pretty large and slow, especially combining three of them altogether. The efficiency comparison, such as memory usage or time required to create synthetic OOD images for training, is not discussed in the paper.

•	Though using a series of Foundational Models can achieve OOD image generations, it could also potentially accumulate their biases or noise. Are there failed cases of generated images?

•	Minor issue:
In Fig. 5, the last row is labeled leaf beetles while the images seem to be keyboard.

**Suitability:**

2

---

### Meta-Review · Area_Chair_QuDJ · 2024-07-02

**Recommendation:** Accept (Poster)
**Confidence:** 5

**Metareview:**

This paper aims to leverage the power of foundation models to address the OOD detection problem. The basic idea is to generate higher-quality outliers to ensure that the model have a better OOD-detection ability. However, the experiments still need improvement, as many outlier-exposure methods are not included in the current version. We encourage merging more baselines' results into the paper in the next round.

TPC addendum: This paper overall has positive reviewer ratings. Reviewers are also generally happy with the rebuttal. Reviewers also raised some issues regarding lack of comparisons with some existing methods. Authors must incorporate these studies into their final version. The TPCs decide to accept this paper.